# Moving Back to the Parental Home in Times of COVID-19: Consequences for Students’ Life Satisfaction

**DOI:** 10.3390/ijerph191710659

**Published:** 2022-08-26

**Authors:** Richard Preetz, Julius Greifenberg, Julika Hülsemann, Andreas Filser

**Affiliations:** 1SOCIUM Research Center on Inequality and Social Policy, University of Bremen, 28359 Bremen, Germany; 2Institute for Social Sciences, University of Oldenburg, 26129 Oldenburg, Germany; 3Research Data Centre (FDZ), Institute for Employment Research, 90478 Nuremberg, Germany

**Keywords:** returning home, COVID-19, life satisfaction, transition to adulthood, boomeranging, living arrangements

## Abstract

Residential independence from parents is a key marker for young adults’ transition to adulthood. Losing this independence by returning to the parental home marks a regression of adult development with negative implications for returnees’ subjective wellbeing. This paper investigates how a return to the parental home during the COVID-19 pandemic affects the life satisfaction of university students. We used nationwide survey data from German university students (N = 913) to analyze differences in life satisfaction for those who did or did not return to their parental homes. Our results revealed two main findings. First, university students who moved back to their parental home reported significantly lower life satisfaction than those who remained living independently. Second, the association between moving back and life satisfaction varied by age. A return to the parental home was more detrimental to older students’ life satisfaction, while students aged 24 or younger did not experience a significant decrease when moving back to the parental home. We discuss the implications of our findings in the context of young adults’ subjective wellbeing during the COVID-19 pandemic.

## 1. Introduction

Since its beginning in early 2020, the COVID-19 pandemic has fundamentally disrupted social life and living conditions. Social distancing guidelines, stay-at-home orders, and public life restrictions to flatten the curve of infections have changed how people work, live, and interact with each other. Studies worldwide report higher levels of depression and anxiety, lower levels of life satisfaction, and increasing rates of economic and financial uncertainty [1,2,3,4,5]. Population-based studies consistently reveal that young adults are affected by higher rates of depression, anxiety, and decreasing life satisfaction than any other age cohort [2,6,7,8,9,10]. University students in particular have seen their daily life routines disrupted by the COVID-19 pandemic [11,12]. Worldwide, educational institutions were closed and universities rapidly shifted from face-to-face to online teaching [13,14]. Studies reveal that these changes induced various adverse effects, experienced by students as intense feelings of instability and uncertainty [15,16]. This resulted in lower levels of subjective wellbeing because their contact and meeting opportunities with supportive peers were limited or lost, and their current employment situations and future career opportunities were more uncertain than ever [14,15,16,17,18,19,20]. The negative impacts of the COVID-19 pandemic on students are related to the disruption of typical developmental tasks associated with the transition to adulthood, including the emotional, residential, and financial independence from the parents, pursuing educational and career goals, and the formation and maintenance of romantic relationships [21].

Leaving the parental home constitutes a key marker in the transition to adulthood [22,23,24]. Establishing your own household promotes autonomy and fosters feelings of independence and competence. Thus, students might experience a return to the parental home and childhood environment as a regression of their identity and individual development [25,26,27]. Previous findings show that economic crises have served as a push factor for returns to the parental home. For instance, the Great Depression of 2008 has been identified as a major disruptor of emerging adults’ transition towards residential independence [28,29,30,31]. Evidence suggests similar consequences from the COVID-19 pandemic: students moved back to their parental homes because of campus closures and financial difficulties. In Germany, the share of students who lived with their parents was stable at around 23% throughout the years prior to the outbreak of the COVID-19 pandemic [32]. A few months into the pandemic, the share increased up to 32%, the largest share in 30 years [32,33]. Previous findings of the consequences of moving back to the parental home for subjective wellbeing suggest a decrease in life satisfaction and mental health for those who moved back [34,35]. However, there is still scarce evidence about the consequences of returning to the parental home due to the circumstances of the COVID-19 pandemic for individuals’ subjective wellbeing. 

## 2. Returning to the Parental Home

Residential independence marks an important milestone for young adults in their transition to adulthood [36]. Around 80% of emerging adults rate residential independence as an important marker for reaching adulthood [37]. Moving out of the parental home is the most influential factor for young adults’ self-perception as full adults [25]. Therefore, a return to the parental home after already having experienced residential independence constitutes a severe disruption in the transition to adulthood.

Given the specific context of the COVID-19 pandemic, university students may be forced to move back to their parental homes because of campus closures, or financial strains and uncertain labor market prospects. Previous studies identified the main reasons for moving back as transitions in other life domains such as finishing higher education degrees, experiencing partnership dissolution, or economic struggles such as becoming unemployed [23,36,38,39,40]. Comparable to former economic recessions, young workers, especially those entering the labor market during such crises, are most affected by higher unemployment rates, lower wages, and more fixed-term contracts [41,42,43,44,45]. Several studies revealed the profound consequences of the pandemic for young adults, especially university students’ economic situations. More than a third of all students lost a job or internship due to pandemics’ economic consequences [15,33,46]. Moreover, students experience worries about their financial situation and future career. In particular, students reported expecting long-lasting impacts on their careers with decreasing probabilities of finding a job and lower earnings later in life [15,16,46]. Economic setbacks force young adults to return to their parental homes. Evidence from previous crises revealed that worsened economic circumstances disrupted young adults’ residential independence and led to higher rates of returning [28,29,30,31,36,39,40].

In general, we expect a decrease in life satisfaction for students who moved back to their parental homes during the pandemic. Most returns will be unintended due to campus closures, the uncertain labor market situation, or COVID-19-related health issues. Students and their parents might struggle to pay rent and other living expenses or tuition fees, resulting in higher rates of returning to the parental home. Such setbacks to independence disrupt the transition to adulthood. Instead of experiencing autonomy and developing their own way of life, students fall back into former childhood roles and are dependent on their parents again. An unexpected return to the parental home increases conflicts and strain in families when students feel that their sense of adult-like independence is threatened [26,47]. Findings showed rising conflicts and disagreements between adult children and parents over rent, bills, private space, and parental monitoring after returning [48,49,50]. 

While most previous studies analyzed reasons for moving back to the parental home, only a few focused on the consequences of returning for subjective wellbeing. Pre-pandemic findings from the U.S. showed an increase in depressive symptoms for young adults who moved back relative to their independently living peers, with the highest rates for those that specified employment problems as the main reasons for returning home [34,35]. Studies that analyze the outcomes of a move or relocation during the pandemic are contradictory. Findings from Italy and the U.S. showed higher levels of anxiety and loneliness for those students who changed their homes, but findings from Canada showed no effects for depression, anxiety, or stress [51,52,53]. However, only two recent studies specifically investigate the consequences of returns to the parental home during the COVID-19 pandemic. For German students from one university, a recent study uncovers longitudinal changes in life satisfaction and mental health before and during the pandemic [54]. Results show a significant decrease in life satisfaction for those students who returned home. Changes in mental health were not associated with returning to the parental home. For the U.S., a study analyzes open questions answered by students from one university who lived at home with their parents during stay-at-home orders [26]. Two-thirds of these students reported a decline in mental health due to fear, stress, and the loss of typical coping resources such as friends or the wider on-campus community. Moreover, students reported the desire to take a break from family members and reconnect with others. They experienced relationship tension and felt satiated by the same people because of too much togetherness and lack of private space. On the contrary, some students (15%) mentioned positive aspects of moving back to their parents. When students’ peer contacts are limited, they might benefit from familial social support in stressful times. This highlights that moving back to the parental home might also have advantages for students. However, given that the majority of students reports negative experiences and results from previous studies also point towards a negative impact, we generally expect the overall consequences for students’ life satisfaction to be negative. This expectation is based on the findings that moving back to the parental home constitutes a disruption in the transition to adulthood.

The main goal of our analysis is to identify whether students who return to their parental home during the COVID-19 pandemic report a lower life satisfaction than students who remained in housing independent of their parents. Additionally, we test whether returning to the parental home is more detrimental for male or female students, older or younger students, or those students who lost their job. Similar to moving back, job loss results in a decline in subjective wellbeing and (mental) health [55,56,57]. Students who moved back to their parental home and lost their jobs may experience multiplicative adverse effects, resulting in a stronger decrease in life satisfaction. Moreover, the impact of losing residential independence on life satisfaction may depend on the duration of living independently. Older students have lived outside the parental home for longer, and thus have established more autonomous daily life routines and social networks than younger students. A setback in autonomy and a sense of adult-like independence may affect older students after their mid-20s harder, leading to increased conflicts with parents and a stronger decrease in life satisfaction [50]. Therefore, older students have been less comfortable returning to the parental home [49]. Finally, young women and men might experience a return to the parental home differently due to diverging parental expectations and internal familial communication [58]. Daughters face higher expectations to help with housework and are more likely to be monitored by their parents in their social and dating life. In sum, daughters report greater difficulties in establishing relationships of equality with parents than sons, which may result in lower life satisfaction [49].

## 3. Materials and Methods

Data for our study come from a nationwide cross-sectional survey of students in Germany. Participants were recruited between 8 September and 8 October 2020, six months after implementing social distancing guidelines in Germany. The study was advertised via Facebook and learning platforms of German universities using a lottery incentive. Data collection relied on an online survey tool. The study was part of a research project on university students’ social connectedness and wellbeing during the COVID-19 pandemic. The survey collected detailed data on students’ living arrangements immediately before the outbreak of the pandemic and a few months later, as well as other demographics and indicators of subjective wellbeing. The time of data collection coincided with the summer break following the summer semester of 2020, during which university instructions had been fully remote. Universities had closed their campuses for curricular and noncurricular activities as part of their pandemic response. At the time of data collection in the late summer of 2020, the numbers of new registered COVID-19 infections in Germany were at a low level.

### 3.1. Sample

In total, 1272 students participated in the survey. We restricted our analytical sample to those students who lived outside the parental home prior to the pandemic (78%, N = 993) and thus were at risk of moving back into their parents’ home. After listwise deleting cases that lacked information in our variables of interest, our final analytical sample comprised information on 913 students aged 18 to 35 years (*M* = 23.94, *SD* = 3.2). Among the students in the analytical sample, more than one-fifth (21.2%) reported having moved back to their parental home. Similar to other studies about COVID-19 and young adults, our analytical sample includes a higher share of female participants (78.8% vs. 21.2%) [26,59]. Due to a low number of cases, we excluded 15 participants who identified their gender as non-binary. Moreover, 36.9% of participants reported losing their job due to the pandemic. Table 1 summarizes the distribution of the variables in our analytical sample.

### 3.2. Measures

Our analyses focus on life satisfaction as the key outcome and dependent variable. Zero-order bivariate correlations between independent variables are presented in Table 2.

*Life satisfaction:* Life satisfaction is based on evaluating one’s own life [60]. It was measured based on a 10-point Likert scale from 1 (very dissatisfied) to 10 (very satisfied). This satisfaction measurement largely corresponds to the Socio-Economic Panel (SOEP) procedure and other representative population surveys. The item question instructed participants to provide an assessment of their overall life satisfaction. 

*Return to parental home (yes/no):* Participants reported where they lived immediately prior to the pandemic in winter term 2019/20 and summer term 2020 after the pandemic started. Based on these responses, we reconstructed whether participants lived with their parents during the data collection (summer 2020) after not having lived with their parents during the previous winter term 2019/20. 

*Employment situation*: Participants reported whether something changed in their employment situation due to the pandemic. Based on this information, we created a dummy variable capturing whether participants’ employment situation was unchanged (0) or if they lost their job in the wake of the pandemic (1).

*Demographic Information:* Participants provided demographic information, including age and gender.

### 3.3. Statistical Approach

We used quantitative research methods and fit a linear ordinary least-squares regression model to investigate the association between students’ life satisfaction and their return to the parental home. Furthermore, we adjusted our model for the control variables described above:(1)$$Y_{i}=b_{0}+b_{1}X_{i}+b_{2}Z_{i}+\varepsilon_{i}$$
where *Y* represents the score in life satisfaction, *b*_0_ is a fixed constant, *X_i_* is a dummy indicator for returning to the parental home, *Z_i_* is a vector of control variables, and *ε_i_* is the error term. In addition to this basic model, we analyzed whether changes in life satisfaction in response to moving back into the parental home are particularly pronounced for students who lost their job or differ by gender or age. To test these differences, we fitted models including an interaction term between the indicator of moving back into the parental home and gender, age, and participants’ employment situation, respectively. Moreover, we included a quadratic term for age to include a potential nonlinear association between age and life satisfaction. In this case, the model is still linear in the parameters but nonlinear in its explanatory variables, and can still be estimated by ordinary least squares regression [61]. Such modeling strategies are common in economics, demography, sociology, or psychology [61,62,63]. To facilitate the interpretation of the results, we present the results from the regression models, including interactions as adjusted predictions. All analyses were performed using R 4.2.1; visualizations and tables were created using the tidyverse and flextable packages [64,65,66].

## 4. Results

Table 3 displays the coefficients from a multivariate linear regression model predicting the life satisfaction of students. As indicated by the first regression coefficient, students who returned to the parental home reported significantly lower life satisfaction compared to students who did not return.

Moreover, we found that the negative association between returning to the parental home and students’ life satisfaction is more pronounced for older participants. Figure 1 displays the adjusted predictions from a linear ordinary least-squares regression model including an interaction term for age and returning to the parental home. As indicated by the 95% confidence intervals, the difference in life satisfaction between students who did and did not return to their parental home is only statistically significant for students older than 24. At younger ages until 24, the difference in life satisfaction between students who did and did not return to their parental home is not statistically significant. Moreover, Figure 1 illustrates that within the group of returnees, older students exhibit lower life satisfaction compared to younger students. Specifically, returnees after their mid-20s report significantly lower life satisfaction than those in their early 20s. In contrast, the age differential does not exist in the group of students who did not return to their parental homes.

Finally, the association between life satisfaction and returning to the parental home does not differ between genders or employment situations in a statistically significant way (Figure 2). Specifically, returning to the parental home is not differently associated with life satisfaction in both male and female students (left panel of Figure 2). Moreover, job loss does not interact with the negative association between life satisfaction and returning to the parental home. Compared to students who did not lose their job, we did not find a different association between life satisfaction and the return to the parental home for students who lost their job during the COVID-19 pandemic (right panel of Figure 2). The detailed regression results for all interaction models are summarized in Appendix A.

## 5. Discussion

The COVID-19 pandemic has changed the daily life routines of university students dramatically. Educational institutions were closed and universities underwent a rapid shift from face-to-face to online teaching, limiting students’ opportunities for peer contact, raising uncertainties about their future educational and employment careers and worries about their own and family’s health. Moreover, the pandemic resulted in a sudden return to the parental home after living independently for a significant share of students. These returns disrupt one of the key developmental tasks of young adults during the transition to adulthood: gaining residential independence from parents. The aim of our study was to analyze how returning to the parental home during the COVID-19 pandemic was related to students’ life satisfaction. Based on the assumption that moving back to the parental home disrupted students’ adult identity and individual development, we used nationwide survey data to investigate if life satisfaction decreased for those students who returned to the parental home.

Our analysis revealed several main findings. Students who returned to the parental home during the COVID-19 pandemic experienced significantly lower life satisfaction than those who remained living independently. This finding is in line with pre-pandemic results that showed higher depression rates for those returning to the parental home [34,35]. Young adults rate residential independence as one of the key markers within the transition to adulthood [22,37]. Leaving the parental home promotes autonomy and fosters feelings of independence and competence. Therefore, returning to the parental home and childhood environment may feel like a regression in adult development. Instead of experiencing autonomy and developing their own way of life, students fall back into childhood roles and are dependent on their parents again [25,26,27]. Qualitative research has shown that after returning to the parental home, daily interactions between adult children and their parents were dominated by negotiations about their adult-like status [49]. Most returnees mentioned that they were still perceived more as a child than adults. Parents were trying to recreate childhood patterns and keep them in a dependent state by establishing behavioral guidelines and limits in several areas of life [49]. Although returnees try to establish a new equal and adult-like relationship with their parents, they feel that their parents still perceive them as children. After experiencing freedom and making their own decisions, they struggle with the loss of autonomy and parental attempts to remain in control over their daily lives when returning home [49]. Additionally, findings from a sample of returned young adults during the pandemic showed that deteriorated mental health was associated with less perceived parental acceptance as an adult and less autonomy [26]. Decreases in life satisfaction may result from these daily struggles, a lack of acceptance, feelings of lost independence, and regression in adult development.

Another explanation for returnees’ lower life satisfaction may result from higher levels of conflict within the family after moving back home. The outbreak of the COVID-19 pandemic increased family stress and put families under pressure to quickly adapt to new daily routines [67,68,69,70]. The pandemic acted as an ambiguous contextual stressor with an unexpected appearance and an unclear ending. Thus, families were challenged by predicting the further progress of the pandemic and preparing for external changes outside their family lives and beyond their control [71,72]. These aspects were particularly dominant at the onset of the pandemic when educational institutions closed, jobs were lost, and the financial situation worsened for students and parents. Thus, most returns to the parental home resulted from external constraints rather than deliberate decisions by young adults. Families had to adapt to these unplanned living arrangements, including potential areas of conflict such as the loss of personal space, autonomy, privacy, and opportunities to have a break from family members [69]. Although parent–child conflicts generally decrease during the transition to adulthood, young adults co-residing with their parents experience higher levels of parent–child conflicts than those living independently [73,74]. Key areas of conflicts and disagreements between returning adult children and parents include financial issues, private space, the share of housework, and parental monitoring [48,49,50]. While a small share of students rated the return to the parental home as a positive experience, the vast majority emphasized the negative or at least ambivalent aspects due to these conflicts or disagreements [26]. Our findings confirm that this general pattern also emerges for returns to the parental home in the wake of the COVID-19 pandemic.

Our second main finding revealed differences in returnees’ life satisfaction by age, but not gender or employment situation. Specifically, a return to the parental home was more detrimental for older students’ life satisfaction than for students aged 24 or younger. Students until 24 did not differ significantly in their life satisfaction depending on whether they moved back to the parental home or remained in housing independent of their parents. However, for students aged 25 or older, returnees reported a significantly lower life satisfaction than those who did not move back. Findings also showed differences within the group of returnees. Life satisfaction was significantly lower for those who moved back to the parental home after the mid-20s compared to those who returned in their early 20s. Unfortunately, we did not have information about the age when students left the parental home. However, in Germany, the mean age for leaving the parental home is 21 for women and 23 for men [38,75]. Returning shortly after moving out is more common and accepted than in older ages after living autonomously for a longer time. Accepting their previous role within the family and a readjustment to the old patterns of within-family interactions and living arrangements may be easier when the time outside the parental household was short. Previous findings confirmed increased conflicts and lower parent–child relationship quality when young adults moved back after their mid-20s [49,50]. Living outside the parental home for a longer time allowed young adults to establish their autonomous daily life routines and social networks, make their own decisions, and develop an adult-like identity. Parent–child negotiations about returnees’ adult-like status and parental attempts to recreate childhood patterns may be much more frustrating for older students who have already lived independently for a longer time. 

A key limitation of our study is the use of cross-sectional data. Our analysis relies on retrospective information on students’ pre-pandemic living arrangements. However, given the relatively short time between the survey and pre-pandemic living arrangement of 7–8 months, students should be able to recall their place of residence correctly. Furthermore, we compared the life satisfaction of those students who moved back into the parental home with those who did not return. Future studies may use longitudinal data, including pre-pandemic levels of life satisfaction and follow-up students’ living trajectories via a within-individual approach to better investigate the causality of the association between moving back and life satisfaction. We investigated the effect of returning to the parental home explicitly for university students. Population-based studies suggested that young adults generally experienced a decline in life satisfaction [2,6,8,10]. Young adults, including students and non-students, were among the most vulnerable groups in the labor market at the onset of the pandemic. Young people are commonly hit hardest by disruptions of education and work-based training, experience increased difficulties when entering the labor market, and are among the first to lose their jobs in economic crises [76,77]. Therefore, young adults could be forced to move back into the parental home regardless of whether they study. Future studies should analyze whether our findings are replicable for non-student young adults. Furthermore, several studies show that non-binary or LGBTQ+ youth experienced unique stressors due to discrimination, victimization, and rejection from their peers, family, and community, resulting in greater odds of poor mental health and wellbeing [78,79,80,81,82]. Due to the small number of non-binary students in our sample, we could not empirically test differences. However, first evidence suggests similar adverse consequences from returning to the parental home for non-binary youth [83]. Finally, more studies on the potential long-term effects on life satisfaction resulting from returns to the parental home are needed. Individuals adapt to major life-course events, and changes in life satisfaction often show a rapid adjustment to baseline levels [84,85,86]. After an initial decrease in life satisfaction, young adults could adjust to their new living arrangement when challenges and conflicts between parents and returnees decline over time. Thus, levels of life satisfaction might readjust after the initial decline in the immediate aftermath of the return to the parental home.

## 6. Conclusions

Our results reveal that a return to the parental home during the COVID-19 pandemic was associated with significantly lower life satisfaction for university students. In particular, moving back to the parental home was more detrimental for students aged 25 or older. These findings highlight the challenges and consequences of returning to the parental home as a regression in adult development. When designing and planning interventions, student psychological and counseling services should take into account the importance of changes in students’ living arrangements for their subjective wellbeing.

## Figures and Tables

**Figure 1 ijerph-19-10659-f001:**
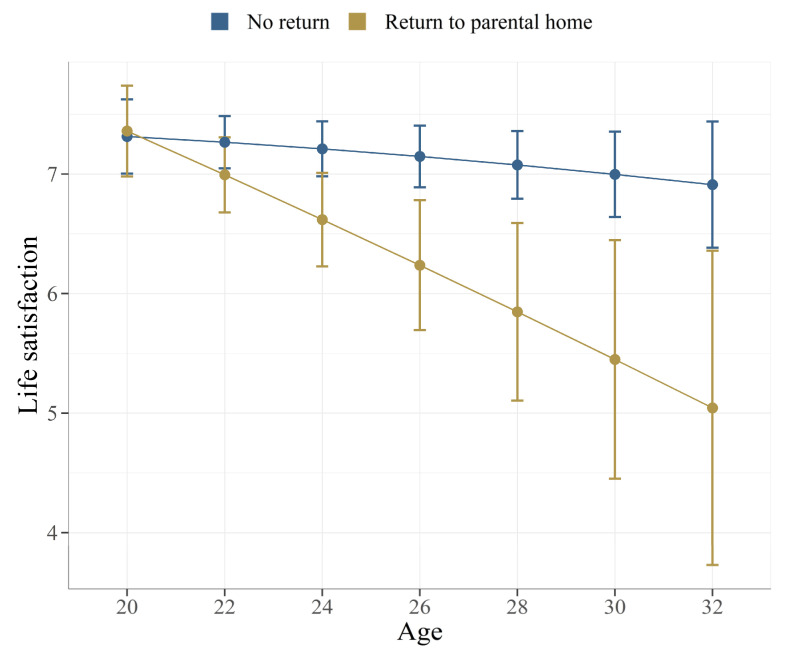
Adjusted predictions of life satisfaction scores by the return to parental home and age of respondents. The model adjusted for gender and employment situation. N = 913.

**Figure 2 ijerph-19-10659-f002:**
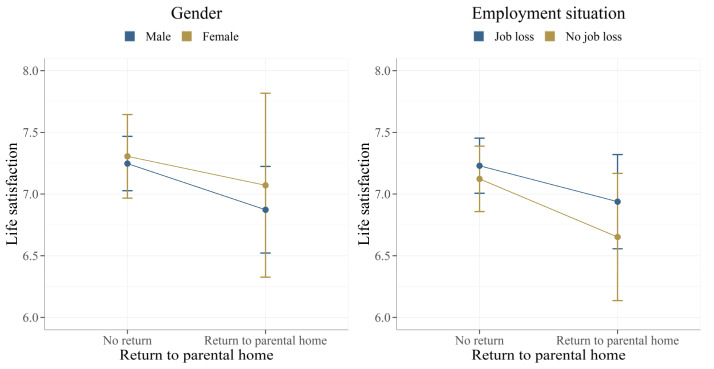
Adjusted predictions of life satisfaction scores by gender and employment situation of respondents. N = 913.

**Table 1 ijerph-19-10659-t001:** Description of the analytical sample.

Variable		n	Mean or %	*SD*	Min	Max
Life satisfaction		913	7.11	2.048	1	10
Return to parental home	No return	719	78.8%			
	Return to parental home	194	21.2%			
Gender	Female	721	79.0%			
	Male	192	21.0%			
Employment situation	No change	576	63.1%			
	Job loss	337	36.9%			
Age		913	23.75	3.207	18	35

**Table 2 ijerph-19-10659-t002:** Correlation matrix.

Variable	Life Satisfaction	Moving Back to Parental Home	Employment Situation	Gender
Return to parental home (ref.: Did not move back)	−0.044			
	[−0.109; 0.021]			
Employment situation: Job loss (ref.: No change)	−0.043	−0.053		
	[−0.107; 0.022]	[−0.118; 0.012]		
Gender Male (ref.: Female)	0.014	−0.071 *	−0.038	
	[−0.051; 0.079]	[−0.135; −0.006]	[−0.103; 0.027]	
Age	−0.071 *	−0.269 ***	0.122 ***	0.086 **
	[−0.135; −0.006]	[−0.328; −0.208]	[0.057; 0.185]	[0.021; 0.15]

Pearson correlation coefficients for all variables in analytical models. Values in square brackets indicate the 95% confidence interval for each correlation. * *p* < 0.05; ** *p* < 0.01; *** *p* < 0.001.

**Table 3 ijerph-19-10659-t003:** Summary of regression analysis of life satisfaction.

Variable		B	SE	min95	max95	*p*
Return to parental home	No return to parental home	ref.				
	Return to parental home	−0.352 *	0.173	−0.692	−0.012	0.042
Employment situation	No change	ref.				
	Job loss	−0.142	0.142	−0.421	0.137	0.319
Age	Age	−0.191	0.251	−0.684	0.301	0.446
	Age^2^	0.003	0.005	−0.007	0.012	0.586
Gender	Female	ref.				
	Male	0.082	0.167	−0.246	0.411	0.624
Intercept		10.205 **	3.120	4.081	16.329	0.001
R^2^		0.011				
adjusted R^2^		0.006				
N		913				

Note. * *p* < 0.05. ** *p* < 0.01.

## Data Availability

Data are available upon request. Please contact the corresponding author.

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
