# Peer review of "Moving Back to the Parental Home in Times of COVID-19: Consequences for Students’ Life Satisfaction"

_ijerph, 2022, doi:10.3390/ijerph191710659_

Round 1

Reviewer 1 Report

 The only observation (and concern) is the rather limited scope of this study; I wonder if this report is is in fact part of a larger study.

Reviewer 2 Report

I applaud the authors for conducting an interesting article with much potential but it will need minor considerations before being published.

The authors repeat the objective of the study in three different parts (Introduction - twice; Returning to the parental home - once). Please consider presenting only in one moment the objective of your study. Usually, the objective is presented just before the methods section.

Methods: How authors determine the number of participants they wanted to include? I can’t find any information about a sample size calculation or information about feasibility considerations. What was your expected power? Please discuss more about it.

Why did you remove the non-binary gender students?  Should not you put this information in the limitations of the study? Is not it important to understand the impact of these students returning home?

I think the tables are not presented in the correct place.

Reviewer 3 Report

See attached file

Round 2

Reviewer 3 Report

In my previous comments I had asked whether all the assumptions of regression were met. When reading the authors' responses they mention that they found collinearity between age and age squared. I had not even noticed in the previous round that age-squared was used in the model, as, in the discussions, the authors only refer to age. I am a qualified statistician and I have a PhD in Mathematical Statistics and I have over 70 publications in accredited international journals, and never in my life have I seen anyone enter a continuous variable in a model and enter a variable into the same model which is just the squared of the variable. The idea of regression models is that the predictors not be highly correlated, because, if they are, then one of them is basically redundant. And this is why one of the assumptions is to check for this and if two predictors are highly correlated to remove one of them. So why would you put the squared value of a predictor into a model when you have the original variable? Statistically, it makes no sense. I'm happy that this article be published if the age_squared variable is removed, but as a specialist in statistics, you can not publish the model as it is.

Author Response

Thank you for your suggestions about our model. Please see the attachment for more information about our modeling strategy.

Round 3

Reviewer 3 Report

Thank you for the clarification. I did some reading and see that adding the squared of a variable is done in some specific fields such as econometrics. As a biostatistician, I've never seen those models being used, however, you have cited literature, they do exist and you could argue for it, so it's OK to keep x-squared in the model. However, as a last comment, I think you should say something about why you've added x-squared in the model for readers such as myself that have not seen this being done. Readers such as myself will think that this leads to multicollinearity and to the use of predictors that are highly correlated (which is not what we want in regression models as it leads to redundant predictors). If you can add a sentence or two at the model motivating why you have included x-squared into your model for your study, the paper may be publishable.

Author Response

Thank you again for your comment and this valuable discussion about this difference between disciplines. These are very good to know, especially when publishing in a multidisciplinary context. We added the following statement in section 3.3 about our statistically approach:

Moreover, we include a quadratic term for age to include a potential non-linear association between age and life satisfaction. In this case, the model is still linear in the parameters but non-linear in its explanatory variables, and can still be estimated by ordinary least squares regression [61]. Such modeling strategies are common in economics, demography, sociology, or psychology [61–63].